# Perceptions of COVID-19 Maternal Vaccination among Pregnant Women and Healthcare Workers and Factors That Influence Vaccine Acceptance: A Cross-Sectional Study in Barcelona, Spain

**DOI:** 10.3390/vaccines10111930

**Published:** 2022-11-15

**Authors:** Elena Marbán-Castro, Ivana Nedic, Mara Ferrari, Esther Crespo-Mirasol, Laia Ferrer, Berta Noya, Anna Marin, Victoria Fumadó, Marta López, Clara Menéndez, Cristina Martínez Bueno, Anna Llupià, Anna Goncé, Azucena Bardají

**Affiliations:** 1ISGlobal, Hospital Clínic, Universitat de Barcelona, 08036 Barcelona, Spain; 2Department of Maternal-Fetal Medicine, BCNatal—Barcelona Center of Maternal-Fetal and Neonatal Medicine, Hospital Clínic and Hospital Sant Joan de Déu, Universitat de Barcelona, 08036 Barcelona, Spain; 3ASSIR Esquerra, Gerència Territorial de Barcelona, Institut Català de la Salut, 08015 Barcelona, Spain; 4Centro de Investigação em Saúde de Manhiça (CISM), Maputo 1929, Mozambique; 5Consorcio de Investigación Biomédica en Red de Epidemiología y Salud Pública (CIBERESP), 28029 Madrid, Spain; 6ASSIR Barcelona Ciutat, Gerència Territorial de Barcelona, Institut Català de la Salut, Universitat de Barcelona, 08007 Barcelona, Spain; 7Department of Preventive Medicine and Epidemiology, Hospital Clinic, Universitat de Barcelona, 08036 Barcelona, Spain

**Keywords:** COVID-19, vaccination, maternal immunisation, vaccine hesitancy

## Abstract

COVID-19 is associated with poor maternal and pregnancy outcomes. COVID-19 vaccination is recommended in Spain, yet vaccination rates in pregnancy are suboptimal. This study investigates the perceptions of pregnant women and healthcare workers (HCW) regarding COVID-19 vaccination. A web-based cross-sectional quantitative study was conducted in 2021–2022 among 302 pregnant women and 309 HCWs in the Catalan public health system. Most pregnant women (83%) and HCWs (86%) were aware of COVID-19 maternal vaccines. The recommendation of the COVID-19 vaccination by an HCW was identified as the greatest facilitator for maternal vaccine uptake, while the fear of harming the foetus was the most significant barrier reported for rejecting vaccination. HCWs recognised they received limited information and training about COVID-19 vaccination in pregnancy, which hindered them from providing informed recommendations. This study highlights that information and education on COVID-19 vaccines to pregnant women and health professionals are pivotal to ensuring informed decision-making and increasing vaccine uptake.

## 1. Introduction

Since the beginning of the COVID-19 pandemic [1], the number of confirmed COVID-19 cases has surpassed 600 million, with more than 6.5 million deaths as of August 2022, according to the World Health Organization (WHO) [2]. The global spread of the COVID-19 pandemic across all regions has led to most pregnancies occurring at this time around the world being at risk of the severe effects of SARS-CoV-2 infection. Accumulated evidence confirms that COVID-19 threatens maternal and perinatal health. Pregnant women are at an increased risk of severe complications (hospitalisations, severe pneumonia, admissions to intensive care units (ICU), invasive mechanical ventilation) and death compared to age-matched non-pregnant women [3]. SARS-CoV-2 infection in pregnancy is associated with an increased risk of having a preterm baby or stillbirth, and neonates born to infected mothers are admitted to ICUs more often than those born to uninfected mothers [4,5,6,7].

The WHO’s first interim recommendations for COVID-19 vaccination under emergency use authorisation for pregnant and lactating women were restricted to healthcare workers (HCW) at a high risk of exposure and women with comorbidities [6,7]. Although recognised as a population of increased risk for severe COVID-19, pregnant women had been excluded from the phase 3 trials of the COVID-19 vaccines and were not recognised as a high-priority group for the allocation of COVID-19 vaccination early in the pandemic [8]. In contrast, in the USA, the Center for Disease Control and Prevention (CDC) advised that pregnant women could be offered COVID-19 vaccinations along with the provision of available information to enable pregnant women to make an informed decision [9]. Later, with the subsequent availability of extensive data from the US from surveillance and population-based cohort studies showing no safety signals among pregnant women who received messenger RNA (mRNA) COVID-19 vaccinations during pregnancy [10,11], as well as the demonstrated high vaccine effectiveness against infection and COVID-19-related complications [9,10,12], recommendations in multiple countries in Europe [13], including Spain [14], were amended to include all pregnant women, advising the use of mRNA vaccines.

Reported differences in COVID-19 vaccination rates between pregnant and non-pregnant cohorts [5,15], with lower vaccination coverage shown among the former group [16], remain an important concern. Evidence on the risk–benefit profile reassures recommendations from scientific societies and regulatory agencies for its use in pregnancy [17,18]. On 11 May 2021, the Government of Spain recommended vaccinating pregnant and lactating women with mRNA vaccines with the rest of the population based on their age and risk group, which was implemented in July 2021 [19]. Given the harmful impact of SARS-CoV-2 infection on maternal and perinatal health [20], understanding the baseline awareness and perceived risks and benefits of COVID-19 vaccination among pregnant women and HCWs is essential for optimal vaccine delivery and the highest impact. 

Vaccine hesitancy, not only in pregnancy, is a complex challenge based on different factors, including region, race, ethnicity, education level, employment status, and social and geopolitical influence [21,22]. A cross-sectional survey performed with 461 pregnant women in Korea showed that the most common reasons for COVID-19 vaccination hesitancy were concerned that the COVID-19 vaccine might affect the foetus (91.7%) and distrust in the vaccine’s effectiveness (42.6%) [23]. HCWs in Turkey shared similar reasons in a survey conducted with 705 HCWs, including doubts about vaccine efficacy, distrust of its content, and fear of its side effects [24].

This study aims to assess the awareness and knowledge among pregnant women and HCWs about COVID-19 maternal vaccination. It also strives to determine the main drivers influencing COVID-19 maternal vaccination acceptability, uptake, and demand by pregnant women and HCWs. 

## 2. Materials and Methods

### 2.1. Study Design and Participants

The data presented here are part of a larger cross-sectional study aimed at understanding the barriers and facilitators to maternal vaccine acceptance among pregnant women and HCWs. This study consisted of a quantitative online survey conducted among pregnant women and HCWs at the Hospital Clínic of Barcelona and Primary Healthcare Centres providing antenatal care (ANC) services in Barcelona, Spain. 

Pregnant women aged between 18 and 55 years, of any gestational age, attending ANC services at Hospital Clínic Barcelona (a tertiary health facility) or Sexual and Reproductive Health Services (Primary Healthcare) in the Barcelona metropolitan area, who are fluent in Spanish or English, were invited to participate in the study. For HCWs, the inclusion criteria included being an HCW providing clinical and/or preventive services to pregnant women in the public health system and being fluent in Catalan or Spanish. Online informed consent was obtained prior to enrolment after the participants were provided with the relevant information about the study purpose in antenatal care visits by an HCW. 

### 2.2. Data Collection Tools

Data collection was obtained through two different questionnaires; one designed specifically for pregnant women and another for HCW. It was conducted anonymously, and a unique identification number was automatically assigned to every participant by the survey platform. The survey covered individual, social, and structural factors that drive vaccine decisions and included constructs related to knowledge, attitudes, beliefs, and practices concerning maternal vaccinations, including COVID-19. The WHO SAGE on Immunization Working Group “Model of determinants of vaccine hesitancy” [25] was used to guide the survey’s development. The survey questions were closed-ended, allowing the participants to choose from a list of predetermined options or to rate their perceptions on a scale. Some questions included the option of “Other”, allowing participants to provide an open answer different from the ones predefined. 

The surveys were administered online through the LimeSurvey data entry system (Version 5.3.6) on the interface of the Hospital Clínic of Barcelona survey platform, *Sistema Enquestes*. Two data collection tools were used, a 42-item questionnaire for pregnant women, which took approximately 10–15 min to complete, and a 21-item questionnaire for HCWs, which could be completed in 5 min. The survey for HCWs (Appendix A) was available in Catalan and Spanish, while the survey for pregnant women (Appendix A) was available in Spanish and English.

The mode of recruitment was both active and passive. For pregnant women, the active recruitment aspect involved HCWs enrolling participants during ANC visits and researchers recruiting participants in waiting rooms. The data collection was then performed either digitally or in paper form. The paper copies were later entered into the survey platform by the study researchers. Regarding the passive recruitment of pregnant women, participants could access the survey via a QR code displayed on posters throughout the health facilities or through fliers provided by HCWs during ANC visits. 

For HCWs, active recruitment involved recruiting study participants through emails containing a survey link, while passive recruitment entailed accessing the survey via a QR code displayed on posters throughout the health facilities. 

### 2.3. Sample Size

The sample size included in the larger study was 611 participants (302 pregnant women and 309 health professionals). These enrolment targets were computed to identify the most conservative response proportion, 50%, using PASS VI 1 (NCSS, LLC; Kaysville, UT, USA). To estimate a response proportion of 50% with a 95% confidence interval of +/− 5% with 80% power, a total of 300 completed surveys were required for each population group.

### 2.4. Statistical Analysis

Data collection, cleaning, and management were performed using the LimeSurvey open source tool through the Hospital Clínic platform. Only fully completed surveys were used for data analysis. For analysis of maternal vaccination coverage, only data from participants eligible for maternal vaccinations at the time of the survey (e.g., week of gestation, season, and COVID-19 recommendations) was included. Categorical data were summarised by frequencies and percentages. Continuous variables were presented as mean and ranges (minimum and maximum values). Missing values were not considered in the description of the data. Data from free-text responses were thematically analysed and manually coded into themes and subthemes to identify the barriers and facilitators for maternal vaccination. Responses provided in Spanish or Catalan were translated into English. For data analysis, all responses were transcribed into the active voice. The resulting responses were organised into codes and sub-codes and displayed in tables (Appendix B and Appendix C).

### 2.5. Ethics Considerations

Ethical approval for this study was granted by the Ethics Review Committees of the Hospital Clínic Barcelona (CEIm) [Reg. No. HCB/2021/0352] and of the Primary Health Care Network Ethics Committee (CEI) IDIAP Jordi Gol [Reg. No. CEI 21/149-P]. The study was conducted in accordance with the Good Clinical Practice Guidelines and under the provisions of the Declaration of Helsinki and national and regional rules and regulations.

## 3. Results

### 3.1. Pregnant Women

Between 1 June 2021 and 24 March 2022, a total of 359 pregnant women were enrolled. Of them, 302 (84.1%) provided complete responses to the survey and were included in the analysis.

#### 3.1.1. Participant Characteristics

The socio-demographic and obstetric characteristics of pregnant women are shown in Table 1. Most women (71.5%) were between 31 and 40 years old. The mean age of women was 34 years old (standard deviation of 4.84). The majority of the women were Spanish (64.6%), held a university or postgraduate degree (74.1%), were employed (80.8%), and were either in a relationship or married (89.8%). Most (73.2%) of the women had a gestational age of more than 24 weeks, and more than half (53.6%) were primigravidae.

#### 3.1.2. Awareness, Views, and Acceptability of COVID-19 Maternal Vaccination

Most pregnant women (82%) reported being aware that maternal vaccinations confer immunological protection to newborns during the first months of life. Of the women surveyed after the national guidelines regarding COVID-19 vaccination were implemented, from 1 August 2021, 83% were aware of the recommendation to be vaccinated against COVID-19 during pregnancy. When asked about their willingness to receive the COVID-19 maternal vaccination during their current pregnancy, 67% indicated they would accept receiving vaccinations. In this period, 56% of participants were vaccinated against COVID-19 (See Figure 1), compared to 7% of the women surveyed before the national recommendations were implemented. The uptake of COVID-19 maternal vaccination increased sharply over the first five months following the introduction of the COVID-19 maternal vaccination recommendation in Spain, though the trend decreased in early 2022 (See Figure 2). In this study, half of the women reported they would take any vaccination recommended by HCWs, whereas 3.6% of women indicated they were not open to receiving any maternal vaccination.

#### 3.1.3. Barriers and Facilitators for COVID-19 Maternal Vaccination

Regarding facilitators for COVID-19 maternal vaccination, the main factor that influenced confidence for vaccine acceptability, reported by 73% of pregnant women, was a recommendation from an HCW (See Figure 3). The wish to protect themselves and their babies from illness was also a significant driver for vaccine acceptability, reported by 54% and 47% of participants who received the COVID-19 vaccine, respectively. About barriers to maternal COVID-19 vaccination reported by mothers who rejected vaccination after the national guidelines were introduced, the fear of the vaccine harming their babies and themselves were reported by 43% and 14% of those who rejected the vaccine, respectively, as the main drivers negatively influencing vaccine uptake (See Figure 4). Not being offered the COVID-19 vaccine was also identified as a barrier to vaccine acceptance by pregnant women. Half of the participants who rejected COVID-19 maternal vaccination indicated, as free-form responses, other factors as barriers. These included the fact of having doubts about the use of novel vaccinations during pregnancy due to the fear of harming the foetus, “*The COVID-19 vaccine is so new that the long-term results it can cause in a developing baby are not known, unlike other vaccines for which there have been studies over the years*” (translated from Spanish; Appendix B), or concerns about how the vaccine would affect the woman during gestation “*Just to state that the only vaccine that I am not in favour of is the COVID one because I think there is still a lot of misinformation and there are still no studies on pregnant women or how it affects women in general hormonally. For me, since I have had fertility problems and would like to get pregnant again, it seems like a risk. An important fact is that I had COVID before pregnancy, and I still have antibodies, which gives me some security*” (translated from Spanish; Appendix B). Feelings of fatigue with the pandemic and the introduction of new vaccinations in general were also reported as influencing vaccine acceptance, “*With COVID-19, I am fed up with vaccines*” (translated from Spanish; Appendix B), as well as a feeling of uncertainty due to conflicting information “*The fact that there is no consensus among doctors is also a source of indecision, we are alone when deciding. I know pregnant women who some doctors have recommended not to be vaccinated against COVID-19, and that generates mistrust*” (translated from Spanish; Appendix B). Moreover, some women expressed that they would receive the vaccination only after pregnancy “*I will get the COVID vaccine once the child is born*” (translated from Spanish; Appendix B). Further details on other reported barriers can be found in Appendix B. 

### 3.2. Healthcare Workers

Between 1 June 2021 and 15 February 2022, a total of 389 HCWs were enrolled. Of them, 309 (79.4%) provided complete responses in the survey and were included in the analysis.

#### 3.2.1. Participant Characteristics

The socio-demographic characteristics of the HCWs who were enrolled are shown in Table 2. The mean age of the HCWs surveyed was 42 years old, ranging from 23 to 65 years. Most HCWs were female (84.8%) and worked as midwives (51.5%) and obstetricians and gynaecologists (35%). Regarding the years of clinical practice, 43.1% had over 15 years of experience, and 30% had between 6 and 15 years. 

#### 3.2.2. Awareness, Views and Perceptions on COVID-19 Maternal Vaccination

Most HCWs (86.4%) participating in the study reported being aware of the recommendation of vaccinating pregnant women against COVID-19. Figure 5 illustrates the views and perceptions of HCWs towards the COVID-19 vaccination in pregnancy, as well as for tetanus, diphtheria and pertussis (Tdap), and influenza maternal vaccines. The data were collected on a five-point scale, ranging from “strongly disagree” (1) to “strongly agree” (5). HCWs showed a high level of confidence in recommending both the Tdap and influenza vaccines, with Tdap being considered the most important vaccine (average of 4.52 on the five-point scale), as well as the safest (4.60) and most effective (4.46). Conversely, HCWs showed considerable uncertainty regarding COVID-19 vaccination, with only 27% of the HCWs strongly agreeing that the vaccine was safe (average of 3.7 points) and 32% reporting the COVID-19 vaccination to be effective (3.89 points) during pregnancy. When the HCWs were asked whether they believed they had sufficient information to recommend maternal vaccines, 24% strongly agreed for COVID-19 (average of 3.48), 40% (4.08) for Tdap, and 33% (3.93) for influenza maternal vaccinations.

The views and perceptions of the HCWs regarding the COVID-19 maternal vaccine (Figure 6) were analysed by comparing the pre- and post-COVID-19 maternal vaccination recommendation periods, from when the vaccine was first recommended by the Government of Spain to pregnant and lactating women, in May 2021. A change in the HCWs’ confidence was observed before and after the COVID-19 maternal recommendation. While in the pre-recommendation period, only 12% of the HCWs strongly agreed that the COVID-19 vaccine was important during pregnancy, this percentage increased to 49% after the recommendation.

The coded qualitative data collected in free-form format on the HCWs’ opinions regarding COVID-19 and other maternal vaccinations is presented in full in Appendix C. The comments pertained to the barriers to maternal vaccination, including the influence of the media and misinformation *“It is very difficult to counteract the effect of the media and fake news regarding vaccination”* (translated from Spanish; Appendix C), the difficulty in changing the opinions of pregnant women who are hesitant to be vaccinated “*I sometimes find it difficult to argue the importance with women who are resistant to vaccines”* (translated from Catalan; Appendix C), and the lack of information that HCWs are faced with, particularly related to COVID-19 maternal vaccination “*I believe that we would need more up-to-date information based on scientific evidence to recommend the COVID19 vaccine to pregnant women”* (translated from Spanish; Appendix C), “*Over the last year, there has been a lack of awareness among team professionals due to a lack of clear direction. You know changes from the news and not from the superiors”* (translated from Catalan; Appendix C). Additionally, many left comments about their perceptions of maternal vaccinations, including their role in maternal vaccination acceptance “*We health professionals need to provide varied, contrasted and evidence-based information. My job is not to recommend, because that would influence the maternal decisions that must be specific to each woman, based on the evidence and her values, priorities, principles, and beliefs”* (translated from Catalan; Appendix C). 

## 4. Discussion

The study aimed to assess the views and perceptions about maternal COVID-19 vaccination among pregnant women and HCWs and to understand the factors that influence COVID-19 maternal vaccination acceptability and uptake. Overall, 302 pregnant women and 309 HCWs were included in the analysis. The findings showed that after the introduction of the COVID-19 vaccination in pregnancy in Spain, 83% of pregnant women and 86% of HCWs were aware of the recommendation. Two-thirds of pregnant women (67%) indicated they would accept receiving vaccinations against COVID-19 during pregnancy, and the vaccine uptake among pregnant women was 56%. These observations were in line with the results of the study by Skirrow et al. [26], who found a 62% willingness to accept COVID-19 vaccination during pregnancy in the UK, significantly higher than the 24% willingness found by Razzaghi et al. in the United States in April 2021 [17]. The uptake of COVID-19 maternal vaccination increased over the first five months following the COVID-19 maternal vaccination national recommendation. At the beginning of the study, only 12% of HCWs reported strongly agreeing that the COVID-19 vaccine was important during pregnancy, yet confidence in COVID-19 maternal vaccination increased up to 50% throughout the duration of the study. 

While the WHO issued interim recommendations for the BNT162b2 Pfizer–BioNTech and mRNA-1273 Moderna COVID-19 vaccines in January 2021 [27,28] and the AZD1222 AstraZeneca COVID-19 vaccine in February 2021 [29] under emergency use authorisation, the vaccination of pregnant women had not yet been a recommendation in Spain at the beginning of this study in March 2021. This led to low confidence in COVID-19 maternal vaccination among the women participating in the study. Following the government recommendation in May 2021 [19] and subsequent vaccine roll-out, the knowledge and acceptance of the vaccination increased among pregnant women and HCWs. Nevertheless, limited information on vaccine safety and misconceptions regarding COVID-19 vaccines [30], as similarly observed in the general population [31], may explain the low (56%) uptake of COVID-19 vaccination in this particular population [32]. A sharp increase in the COVID-19 maternal vaccine coverage among pregnant women was observed following the effective roll-out of vaccines, with a decrease in vaccination uptake in December 2021, similar to that observed in other population groups [33]. While it is difficult to discern the reasons for this decline, with the recommendation for the third booster dose in the general population, a rise in vaccine uptake among pregnant women was later observed in February 2022. 

Regarding factors acting as major decision-makers to receive a vaccination, a recommendation from HCWs showed the most pivotal influence for pregnant women, with 73% naming it as an influencing factor. This finding mirrors other studies on maternal vaccines other than COVID-19 [34] and highlights the crucial role of HCWs regarding the information, education, and recommendation of maternal vaccinations during ANC appointments. Following this, the wish to protect themselves from illness (54.3%) and their babies from illness (46.6%) were seen as leading facilitators of vaccination. Based on the qualitative data, it was also found that the women believed that receiving vaccinations was the only way to end the pandemic and feared being unable to take medication if they were infected by the virus. Conversely, for the maternal COVID-19 vaccination, the fear of harming the baby was a leading barrier, with 42.5% of women not vaccinated stating that it was a barrier, even after the government officially recommended the vaccination. Based on the qualitative data collected, many women also reported the limited data and the uncertainty regarding studies about the new vaccine, the lack of clinical trial data including pregnant women, and the abundance of vaccine misinformation. Additionally, 13.8% of women indicated that they were not offered the vaccine by HCWs. This may have been due to their gestational age, with some HCWs reporting avoiding recommending the vaccination in the first and third trimesters, according to our qualitative data. 

In line with previous studies [35], our results show a difference in the views and perceptions about the COVID-19 maternal vaccination of HCWs before and after the government recommendation, with over 80% of them reporting that the COVID-19 vaccination was important during pregnancy after the recommendation, compared to 33% prior. This mirrors other studies conducted among HCWs in Spain, such as that of Palma et al. [36], showing a decrease in vaccine hesitancy among the HCWs surveyed between 2020 and 2021. Interestingly, this was also observed in maternal vaccination uptake, where 7% of the women surveyed before the national recommendations reported having been vaccinated against COVID-19 while pregnant, increasing to nearly 60% after the national vaccine recommendation in pregnancy. Nevertheless, while most HCWs considered COVID-19 an important maternal vaccine, their trust in the Tdap and influenza vaccinations remained higher, and they reported having more information about these vaccinations.

The semi-qualitative approach of this study, through some free-form questions, enabled the study participants to express views on maternal vaccinations that were not investigated through the survey. Certain reoccurring themes were noted, such as the trust in the HCW recommendations and the fear of becoming infected with COVID-19 during pregnancy, as no specific treatment was perceived as being available for the treatment of COVID-19 in pregnancy at the time, which were shown as facilitators to vaccination. Remarkably, this study also revealed that while the majority (83%) of HCWs believed that the recommended COVID-19 maternal vaccination was important for the protection of pregnant mothers, less than a third recognised that they had sufficient information regarding the COVID-19 maternal vaccine. Proving the key role that health professionals play in vaccine acceptability, the uptake and demand of COVID-19 maternal vaccination should translate into prioritisation, specific training, and information provided to HCWs on maternal vaccines for higher impact, as also shown in other studies [36]. A reoccurring barrier to COVID-19 vaccine coverage was the lack of COVID-19 vaccine trial data in pregnant women, as outlined in the qualitative aspect of this study. Pregnant women are rarely represented in clinical trials due to ethical issues, with ongoing debates on whether they should be included [37]. 

This study has certain limitations, some related to quantitative study design. Namely, the use of closed-ended questions throughout the survey, with specific options to choose from and little or no opportunity for explanations. While there was the option of “Other” with the possibility of a free-form response, the existence of predetermined answer options could have caused a disposition to select a listed option. Additionally, the surveys for pregnant women were only available in Spanish and English, meaning that there may have been a language barrier for some participants or that certain potential participants could have been excluded. In terms of the study sample, most surveys for pregnant women were conducted in two centres, the Hospital Clínic Barcelona and the CAP Manso, the former receiving more high-risk pregnancies and the latter serving pregnancies from the local community. As the study sample represents the population of Barcelona, this could have led to generalisations. Additionally, the surveys did not include relatives involved in the decision-making process, such as women’s partners. Finally, though the calculated sample size of the larger study was 300 pregnant women and 300 HCWs, cleaning the data to only include those interviewed after the national guidelines were introduced led to smaller sample sizes for both groups being included in this manuscript.

## 5. Conclusions

This cross-sectional study provided insights into the perceptions of pregnant women and HCWs regarding COVID-19 vaccination during pregnancy and the barriers and facilitators to vaccine uptake. The COVID-19 vaccination in pregnancy was recommended in Spain during the conduction of the study, and awareness of the vaccination was relatively high among both groups. The study demonstrated that HCWs are the greatest factor in decision-making and hold a key role in COVID-19 maternal vaccine acceptance, with their recommendation resulting in increased coverage. The majority of HCWs self-reported recommending maternal vaccinations, yet they recognised a lack of information and training regarding COVID-19 vaccinations. The complex and interrelated factors affecting maternal vaccination acceptance and uptake need to be addressed when optimising service delivery strategies to increase acceptability of the intervention and to achieve a high vaccination coverage. 

## Figures and Tables

**Figure 1 vaccines-10-01930-f001:**
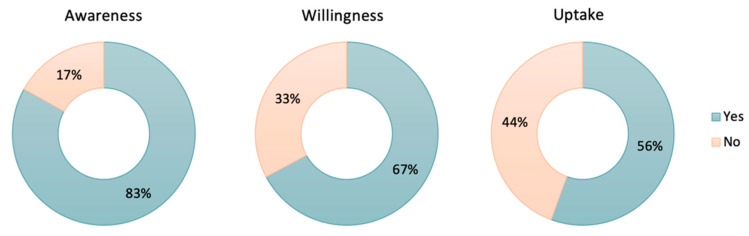
Awareness, willingness, and uptake of COVID-19 maternal vaccination among pregnant women eligible for vaccination during current pregnancy. Includes data only for the study period following the effective implementation of the maternal COVID-19 national recommendation, from 1 August 2021.

**Figure 2 vaccines-10-01930-f002:**
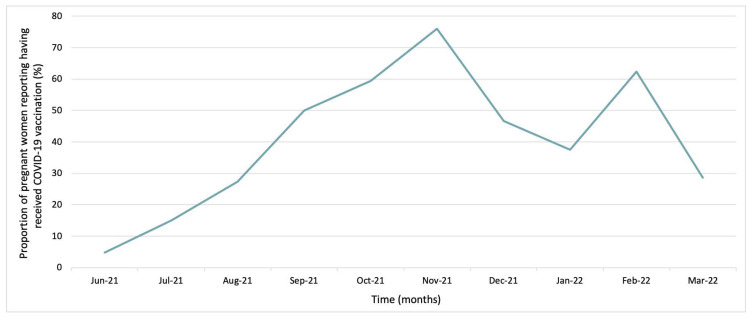
Trends of COVID-19 maternal vaccination uptake after the introduction of national recommendation.

**Figure 3 vaccines-10-01930-f003:**
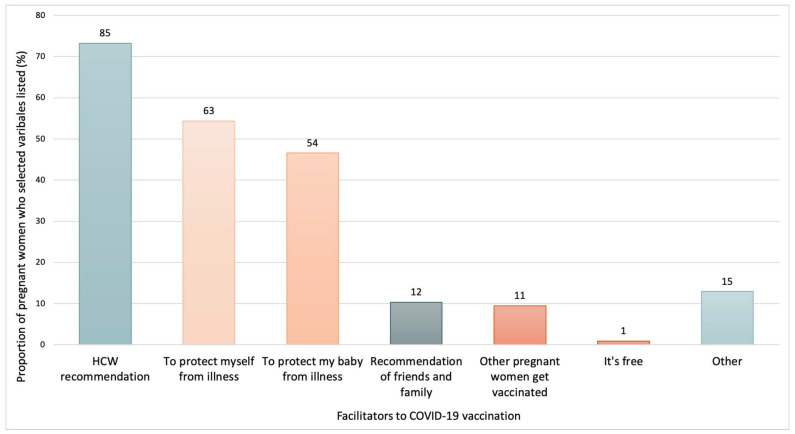
Facilitators for acceptance of COVID-19 vaccination reported by the pregnant women who received the maternal COVID-19 vaccination. N = 116. More than one answer possible. The figure shows data from participants enrolled during the period of the study after the effective implementation of national maternal COVID-19 recommendations, from 1 August 2021.

**Figure 4 vaccines-10-01930-f004:**
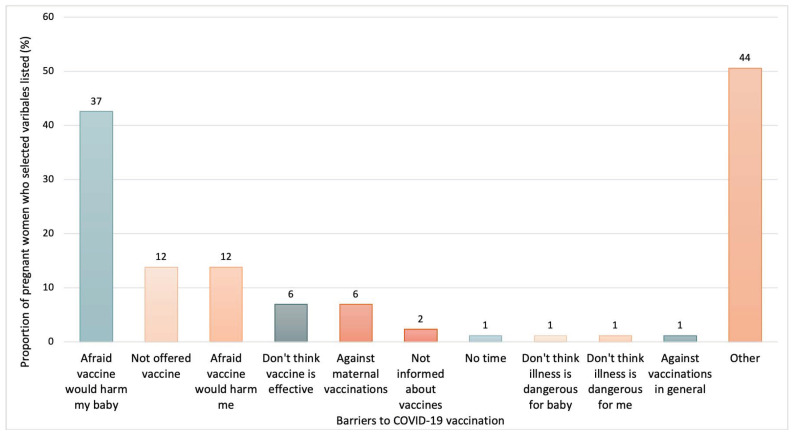
Barriers to vaccinating against COVID-19 during the current pregnancy reported by the pregnant women who rejected the maternal COVID-19 vaccination, N = 87. More than one answer possible. The figure shows data from participants enrolled during the period of the study after the effective implementation of national maternal COVID-19 recommendations, from 1 August 2021. No women selected the option ‘Religion or cultural beliefs’.

**Figure 5 vaccines-10-01930-f005:**
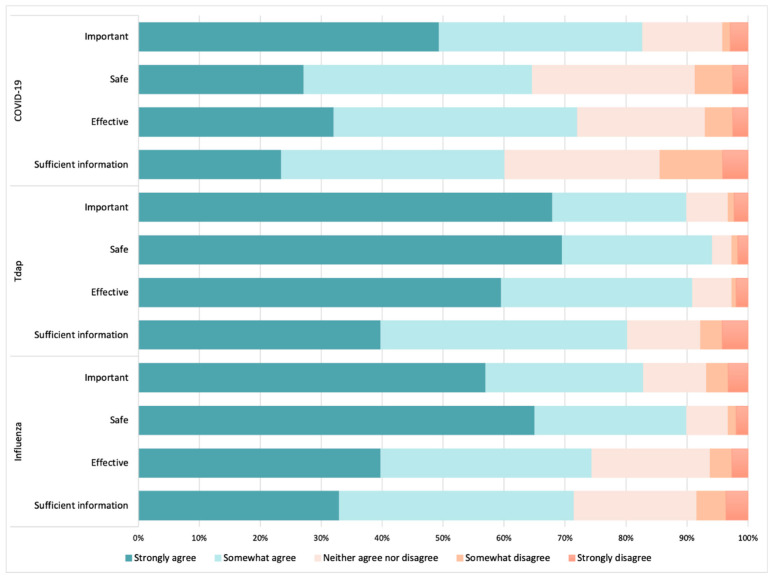
Views and perceptions of HCWs towards COVID-19 vaccination in pregnancy as well as other maternal vaccinations. The data regarding the COVID-19 vaccine above only include data collected after the guidelines for this vaccine for pregnant women came into practice in Spain after July 2021. N COVID-19 = 196; N Tdap = 302; N Influenza = 302.

**Figure 6 vaccines-10-01930-f006:**
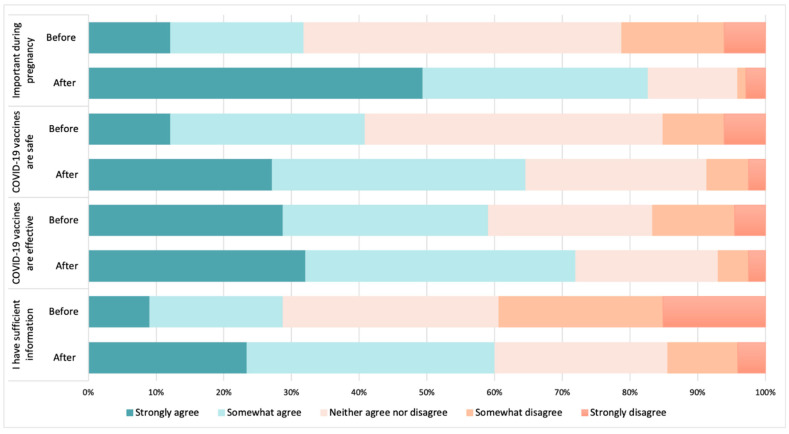
Views and perceptions of HCWs towards COVID-19 vaccination during pregnancy and other maternal vaccinations, before and after COVID-19 maternal vaccination national guidelines in May 2021.

**Table 1 vaccines-10-01930-t001:** Socio-demographic and obstetric characteristics of pregnant women at enrolment.

	Pregnant Women (N = 302)	
Characteristics	n	(%)
**Age** (years)		
≤25	16	(5.3)
26–30	45	(14.9)
31–35	112	(37.1)
36–40	104	(34.4)
41–45	25	(8.3)
Mean (SD)	34 (4.84)	
**Nationality**		
Spain	195	(64.6)
Argentina	9	(3.0)
Venezuela	11	(3.6)
Italy	10	(3.3)
Pakistan	7	(2.3)
Brazil	5	(1.7)
Ecuador	6	(2.0)
France	4	(1.3)
Russia	4	(1.3)
The Dominican Republic	5	(1.7)
Peru	5	(1.7)
Other countries ^1^	41	(13.6)
**Education**		
Primary education	5	(1.7)
Secondary education	32	(10.6)
Vocational training	41	(13.6)
University degree	104	(34.4)
Postgraduate studies	120	(39.7)
**Occupation**		
Employed	210	(69.5)
Self-employed	34	(11.3)
Unemployed	54	(17.9)
Student	4	(1.3)
**Gestational age (weeks)**		
<12	10	(3.3)
12–23	71	(23.5)
24 or more	221	(73.2)
**Marital status**		
Single	29	(9.6)
In a relationship	137	(45.4)
Married	134	(44.4)
Divorced	2	(0.7)
**Gravidity**		
Primigravidae	162	(53.6)
Multigravidae	140	(46.4)

^1^ Other countries: Albania, Andorra, Bangladesh, Bolivia, Canada, Chile, Colombia, Cuba, El Salvador, U.S.A., Philippines, Greece, Equatorial Guinea, Honduras, India, Morocco, Mexico, Nigeria, Paraguay, Poland, Portugal, United Kingdom, Western Sahara, Ukraine, Uruguay.

**Table 2 vaccines-10-01930-t002:** Socio-demographic characteristics of HCWs participating in the study.

	HCW (N = 309)	
Characteristics	n	(%)
**Age category (years) ***		
≤30	65	(21.7)
31–50	156	(50.5)
≥51	86	(27.8)
Mean age in years (SD)	42 (12.19)	
**Gender**		
Female	262	(84.8)
Male	44	(14.2)
Undisclosed	3	(1.0)
**Nationality**		
Spain	285	(92.2)
Argentina	6	(1.9)
Italy	3	(1.0)
Peru	4	(1.3)
Venezuela	3	(1.0)
Other ^1^	8	(2.6)
**Profession**		
Obstetrician/Gynaecologist	108	(35.0)
Midwife	159	(51.5)
Nurse	18	(5.8)
Family doctor	7	(2.2)
Paediatrician	6	(1.9)
Neonatologist	11	(3.6)
**Clinical practice (years)**		
1–5	81	(26.2)
6–10	48	(15.5)
11–15	47	(15.2)
>15	133	(43.1)

^1^ Other countries: Bolivia, Canada, Colombia, Dominican Republic, Ecuador, and Morocco. * Three missing values.

## Data Availability

The data presented in this study are available on request from the corresponding author. The data are not publicly available due to confidentiality issues with the individuals surveyed.

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
