# Peer review of "Perceptions of COVID-19 Maternal Vaccination among Pregnant Women and Healthcare Workers and Factors That Influence Vaccine Acceptance: A Cross-Sectional Study in Barcelona, Spain"

_vaccines, 2022, doi:10.3390/vaccines10111930_

Round 1

Reviewer 1 Report

The article is well structured and interesting.
However, I suggest implementing the introduction with more background analysis of vaccine hesitancy. In particular, it could be useful to describe the reasons that led pregnant women (like other subjects) to have doubts about the effectiveness of vaccination. It might be useful to retrace the vaccine production process and the various discussions that have taken place in the literature. Then the authors could emphasize the efficacy of the vaccine more based on the comparison of existing literature.

Both to briefly explain the path that led to the synthesis of an effective vaccine, and to confirm the effectiveness of the vaccine (and any 'problems') I suggest, for example, these articles:

- Knoll MD, Wonodi C. Oxford-AstraZeneca COVID-19 vaccine efficacy. Lancet. 2021 Jan 9;397(10269):72-74. doi: 10.1016/S0140-6736(20)32623-4. Epub 2020 Dec 8. PMID: 33306990; PMCID: PMC7832220.

- Ciotti M, Ciccozzi M, Pieri M, Bernardini S. The COVID-19 pandemic: viral variants and vaccine efficacy. Crit Rev Clin Lab Sci. 2022 Jan;59(1):66-75. doi: 10.1080/10408363.2021.1979462. Epub 2021 Oct 1. PMID: 34598660.

- Cioffi A. Coronavirus Disease 2019: Is Everything Lawful to Create an Effective Vaccine? J Infect Dis. 2020 Jun 16;222(1):169. doi: 10.1093/infdis/jiaa216. PMID: 32348489; PMCID: PMC7197522.

- Luxi N, Giovanazzi A, Capuano A, Crisafulli S, Cutroneo PM, Fantini MP, Ferrajolo C, Moretti U, Poluzzi E, Raschi E, Ravaldi C, Reno C, Tuccori M, Vannacci A, Zanoni G, Trifirò G; Ilmiovaccino COVID19 collaborating group. COVID-19 Vaccination in Pregnancy, Paediatrics, Immunocompromised Patients, and Persons with History of Allergy or Prior SARS-CoV-2 Infection: Overview of Current Recommendations and Pre- and Post-Marketing Evidence for Vaccine Efficacy and Safety. Drug Saf. 2021 Dec;44(12):1247-1269. doi: 10.1007/s40264-021-01131-6. Epub 2021 Nov 5. PMID: 34739716; PMCID: PMC8569292.

- Garg I, Shekhar R, Sheikh AB, Pal S. COVID-19 Vaccine in Pregnant and Lactating Women: A Review of Existing Evidence and Practice Guidelines. Infect Dis Rep. 2021 Jul 31;13(3):685-699. doi: 10.3390/idr13030064. PMID: 34449637; PMCID: PMC8395843.

- Yoon H, Choi BY, Seong WJ, Cho GJ, Na S, Jung YM, Jo JH, Ko HS, Park JS. COVID-19 Vaccine Acceptance during Pregnancy and Influencing Factors in South Korea. J Clin Med. 2022 Sep 28;11(19):5733. doi: 10.3390/jcm11195733. PMID: 36233601; PMCID: PMC9573627.

Piekos SN, Price ND, Hood L, Hadlock JJ. The impact of maternal SARS-CoV-2 infection and COVID-19 vaccination on maternal-fetal outcomes. Reprod Toxicol. 2022 Oct 22:S0890-6238(22)00153-8. doi: 10.1016/j.reprotox.2022.10.003. Epub ahead of print. PMID: 36283657; PMCID: PMC9595355.

Author Response

Thank  you very much for providing such a detail list of references. We have improved the manuscript taking into account the articles provided. We hope that the new version meets your expectations.

Reviewer 2 Report

In this manuscript, the author describe how did they run a survey to know which were the perceptions of a group of pregnant women and healthcare workers working with this population. The text is interesting and well-written, however, there are some aspect that need special attention. Nevertheless, congratulations to the authors.

ABSTRACT/TITLE

- I would suggest to include the time of study, in order to give a better context to the whole text.

INTRODUCTION

- Since the beginning of the "COVID-19" pandemic.

- Which date are these prevalence numbers from?

- "currently available COVID-19 vaccines": could you list which? (at least in your geographical setting).

- Please, include the meaning of all the abbreviations the first time you mention them: mRNA (for instance).

- I would suggest to use the term "healthcare worker (HCW)".

METHODS

- "The data presented here is part of a larger cross-sectional study" Is there a reference for that?

- "who are fluent in Spanish or English" No interview was done in Catalan?

- "included being a health professional" If you already included an abbreviation for this, I would suggest to use it comprehensively.

- Before proceeding further with continuous variables, I would suggest to check their distribution. Was this done? If the distribution is non-parametric, I would suggest to use median and interquartile range.

RESULTS

- Could it be possible to provide the n of pregnant women per country underneath the table, together with the country listing? (both pregnant women and HCW).

- Please, included the meaning of "tdap".

DISCUSSION

- Bias: survey completion before Spanish Govt. recommendation for vaccination. What about govts. from origin countries in respondents of foreign origin? Could that be relevant? And regarding the Regional Catalan Govt., was there a consensus between the Spanish and the Catalan Govt. with this regards (as the Health policies are of the regions in Spain).

FIGURES

- Why did the vaccination trend got reduced after November 2021?

- Please, include in all the graphics the meaning of the colour coding. I suggest to keep a coherence in the colour coding throughout all the manuscript not to confuse the reader.

APPENDIX

- There are several orthographic and grammatical mistakes in the Spanish version (lack of accents, of certain letters (ñ), exclamation marks at the beginning and end of the exclamation, "Dios/God" [with capital letter]...).

OTHER

- Could it be possible to provide the respective surveys as appendix?

- Were HCW asked whether they would recommend the vaccine or not when answering the questionnaire?

- Could you describe how did the participants get aware of your survey? How were they approached?

- Was there any correlation between accepting/rejecting other vaccines and the COVID-19 one in either group?

- The HCW from this answer were somehow in charge of the pregnant women included to analysis?

- Does the gender of the HCW have any influence in the potential behaviour of the pregnant?

- Was influencing in any way the type of vaccine to be administered (mRNA, vector-based)

- I miss more discussion about the relevant to get answers both from pregnant and HCW. Of course it´s relevant, because one can influence the other, but I wonder whether you could make any correlation, comparison, etc. Maybe this and some of my concerns above are described in the bigger publication :)

Author Response

Reviewer 2: In this manuscript, the author describe how did they run a survey to know which were the perceptions of a group of pregnant women and healthcare workers working with this population. The text is interesting and well-written, however, there are some aspect that need special attention. Nevertheless, congratulations to the authors.

Authors’ response: Thank you very much for your constructive feedback.

ABSTRACT/TITLE

- I would suggest to include the time of study, in order to give a better context to the whole text.

Authors’ response: Thank  you very much for this suggestion. Due to the length of the title, we have instead included the timeframe in the abstract, to give more context.

INTRODUCTION

- Since the beginning of the "COVID-19" pandemic.

Authors’ response: Thank you, the word “COVID-19” has been added to the sentence.

- Which date are these prevalence numbers from?

Authors’ response: The date of the reference has been added (August 2022).

- "currently available COVID-19 vaccines": could you list which? (at least in your geographical setting).

Authors’ response: Thank you for highlighting this. We have removed the words “currently available” as it might be redundant.

- Please, include the meaning of all the abbreviations the first time you mention them: mRNA (for instance).

Authors’ response: Thank you for pointing this out. We have added the meanings of abbreviations such as mRNA, CDC and WHO.

- I would suggest to use the term "healthcare worker (HCW)".

Authors’ response: Thank you for the suggestion. We have changed the wording throughout the manuscript.

METHODS

- "The data presented here is part of a larger cross-sectional study" Is there a reference for that?

Authors’ response: The manuscript describing the larger study is pending to be submitted for publication shortly. Unfortunately, we cannot add a reference yet.

- "who are fluent in Spanish or English" No interview was done in Catalan?

Authors’ response: No, the survey for pregnant women was only available in Spanish and English. The healthcare workers had the option to answer the survey in Spanish and Catalan.

- "included being a health professional" If you already included an abbreviation for this, I would suggest to use it comprehensively.

Authors’ response: Thank you, the “HCW” abbreviation has been added throughout the manuscript.

- Before proceeding further with continuous variables, I would suggest to check their distribution. Was this done? If the distribution is non-parametric, I would suggest to use median and interquartile range.

Authors’ response: Thank you for suggesting this. The continuous variables are normally distributed.

RESULTS

- Could it be possible to provide the n of pregnant women per country underneath the table, together with the country listing? (both pregnant women and HCW).

Authors’ response: Both the “n” and “N” are provided for pregnant women and HCW in the tables. Could you clarify which “n” should we provide? Thanks for the clarification.

- Please, included the meaning of "tdap".

Authors’ response: The meaning of the acronym has been added (line 251).

DISCUSSION

- Bias: survey completion before Spanish Govt. recommendation for vaccination. What about govts. from origin countries in respondents of foreign origin? Could that be relevant? And regarding the Regional Catalan Govt., was there a consensus between the Spanish and the Catalan Govt. with this regard (as the Health policies are of the regions in Spain).

Authors’ response: All pregnant women, regardless of their country of origin, were following their pregnancies at an antenatal care facility in Catalonia (as per inclusion criteria). Thus, this potential bias is minimized as

FIGURES

- Why did the vaccination trend got reduced after November 2021?

Authors’ response: Unfortunately, we have no data to explain this reduction in uptake. In the discussion we mentioned the following potential explanation: “A sharp increase in COVID-19 maternal vaccine coverage among pregnant women was observed following the effective roll-out of vaccines, with a decrease in vaccination uptake in December 2021, similar to that observed in other population groups [27]. While it is difficult to discern the reasons for this decline, with the recommendation for the third booster dose in the general population, a rise in vaccine uptake among pregnant women was later observed in February 2022”.

- Please, include in all the graphics the meaning of the colour coding. I suggest to keep a coherence in the colour coding throughout all the manuscript not to confuse the reader.

Authors’ response: Thank you very much for highlighting this. We have changed the only two figures that could have a grading colour scale (from strongly agree to strongly disagree), to facilitate reading.

APPENDIX

- There are several orthographic and grammatical mistakes in the Spanish version (lack of accents, of certain letters (ñ), exclamation marks at the beginning and end of the exclamation, "Dios/God" [with capital letter]...).

Authors’ response: Thank you for highlighting this. We copied-pasted participants’ responses to keep their narratives as similar as possible to their own language. But we agree that grammatical errors could have been edited. We have reviewed the sentences to be grammatically correct.

OTHER

- Could it be possible to provide the respective surveys as appendix?

Authors’ response: Yes, we agree that they should be provided. The surveys have been provided in the supplementary material, for reference.

- Were HCW asked whether they would recommend the vaccine or not when answering the questionnaire?

Authors’ response: Yes, but this was asked in general (whether they would recommend maternal vaccinations), not specifically about COVID-19 vaccination. We are currently drafting another manuscript with the data from the whole quantitative study (on maternal vaccination in general; not vaccine specific) and will include that data there.

However, if you mean that if HCW knew that they were going to be asked about their recommendations before accepting participation, the answer is no. We understand that it might have biased participation.

- Could you describe how did the participants get aware of your survey? How were they approached?

Authors’ response:  Thank you for the suggestion. We have added the following in page 3, section “Data collection tools”, line 119: “The mode of recruitment was both active and passive. For pregnant women, the active recruitment aspect involved HCW enrolling participants during antenatal visits and researchers recruiting the participants in waiting rooms. The data collection was then performed either digitally or in paper form. The paper copies were later entered into the survey platform by the study researchers. Regarding the passive recruitment of pregnant women, participants could access the survey via a QR code displayed on posters throughout the health facilities or through fliers provided by HCW during antenatal visits.

For HCW, active recruitment involved recruiting study participants through emails containing a survey link, while passive recruitment entailed accessing the survey via a QR code displayed on posters throughout the health facilities.

- Was there any correlation between accepting/rejecting other vaccines and the COVID-19 one in either group?

Authors’ response: A potential correlation will be analysed and discussed in another manuscript that we are currently drafting, relating to all three currently recommended maternal vaccinations. This future manuscript will be a quantitative study covering all aspects of the survey in question in this paper.

- The HCW from this answer were somehow in charge of the pregnant women included to analysis?

Authors’ response: We unfortunately don’t have data on whether the HCW were working with the specific pregnant women surveyed, however, they were recruited in the same locations.

- Does the gender of the HCW have any influence in the potential behaviour of the pregnant?

Authors’ response: This is a very interesting question to potentially study. Unfortunately, most of the HCW surveyed (85%) were females, coinciding with the general percentages of female-male HCW ratios, especially at antenatal care visits. So, this question might be difficult to be researched. Qualitative methods could be applied to further explore it.

- Was influencing in any way the type of vaccine to be administered (mRNA, vector-based)

Authors’ response: This is a very interesting question; however it might be out of our research reach. As we did not discuss the vaccine types with the pregnant women, it is difficult to know whether they were aware that different vaccine types exist and which ones they were offered. Therefore, we cannot comment on whether this influenced their acceptance of the vaccine. Additionally, pregnant women would have been offered the mRNA vaccines during pregnancy, so there wouldn’t have been variations in vaccine type (no live virus vaccines could be administered to pregnant women).

- I miss more discussion about the relevant to get answers both from pregnant and HCW. Of course it´s relevant, because one can influence the other, but I wonder whether you could make any correlation, comparison, etc. Maybe this and some of my concerns above are described in the bigger publication :)

Authors’ response: We considered that the sample size was not big enough to make correlations. For that reason, we preferred to display basic demographics.

Thank you very much for such a detailed review of our manuscript. Your comments have helped us improved the quality of our publication. We hope that the new version meets your expectations.

Reviewer 3 Report

Authors conducted a study to find out perceptions of pregnant women and healthcare professionals dealing with pregnant females about COVID-19 vaccination. The introduction section is clearly written with proper background information, literature citation and a definite need of current study. However there are few missing points in methodology section. When data was conducted anonymously, how a unique tracking number was assigned. Moreover information regarding participants recruitment is missing. The methodology section should clearly elaborate how participants were approached, how the study purpose was explained and procedure of filling of study instrument, in current case its online so it needs more detailed information about how contacts of participants were retrieved, were they given an option to decline to participate in study, if yes how it was given.The enrollment period is around 9 months, so how participants were explained about study procedure. The methodology section is very confusing and needs attention.

Line no 192 "Not being offered the COVID-19 vaccine was also identified as a barrier to vaccine acceptance by pregnant 193 women". Explain the reason why pregnant females were not offered vaccination despite recommendation.

How was the english and spanish survey similarity validated. In case of self made survey, what was the realibility and validity of questionnaire.

The authors should explain how they analyze the qualitative response regarding barriers of COVID-19 vaccination, since there were hundreds of pregnant females, how did they perform thematic analysis of their responses.

For healthcare professionals, maximum data (50%) is collected from mid-wifes. This will create bias, as the level of awareness of a specialist, gynaecologist will not be same as a mid wife. Authors should keep this factor in mind while explaining results.

Discussion and conclusion are well written.

Author Response

Authors conducted a study to find out perceptions of pregnant women and healthcare professionals dealing with pregnant females about COVID-19 vaccination. The introduction section is clearly written with proper background information, literature citation and a definite need of current study.

Authors’ response: Thank you for this feedback.

However there are few missing points in methodology section. When data was conducted anonymously, how a unique tracking number was assigned. Moreover information regarding participants recruitment is missing. The methodology section should clearly elaborate how participants were approached, how the study purpose was explained and procedure of filling of study instrument, in current case its online so it needs more detailed information about how contacts of participants were retrieved, were they given an option to decline to participate in study, if yes how it was given. The enrollment period is around 9 months, so how participants were explained about study procedure. The methodology section is very confusing and needs attention.

Authors’ response: Thank you for your comments. We have added the required information.

Please find information about how the identification number was assigned in page 3, section Data collection tools, lines 103-104: “a unique identification number was automatically assigned to every participant by the survey platform”.

Information on how the participants were recruited is detailed in page 3, section Data collection tools, lines 119-129.

Information about how the purpose of the study has been explain in page 3, section Study design and participants, lines 98-99: “after the participants were provided with the relevant information about the study purpose in antenatal care visits by a HCW”.

Line no 192 "Not being offered the COVID-19 vaccine was also identified as a barrier to vaccine acceptance by pregnant 193 women". Explain the reason why pregnant females were not offered vaccination despite recommendation.

Authors’ response: We have added an explanation in the Discussion (line 433). Some HCW stated that they avoid recommending the COVID-19 vaccine to pregnant women during the first and last trimester of pregnancy.

How was the english and spanish survey similarity validated. In case of self made survey, what was the realibility and validity of questionnaire.

Authors’ response: Both surveys were designed for this purpose and internally validated by the research team to ensure that the questions effectively capture the topic under investigation. The research team are fluent in both languages, so both surveys could be validated in parallel, also ensuring readability.

The authors should explain how they analyze the qualitative response regarding barriers of COVID-19 vaccination, since there were hundreds of pregnant females, how did they perform thematic analysis of their responses.

Authors’ response: Thematic analysis was performed with the free-form (qualitative) responses given by pregnant women and HCW, by manually coding the answers into themes and subthemes. We have clarified this in the manuscript (line 169).

For healthcare professionals, maximum data (50%) is collected from mid-wifes. This will create bias, as the level of awareness of a specialist, gynaecologist will not be same as a mid wife. Authors should keep this factor in mind while explaining results.

Authors’ response: We are aware of this potential bias. We tried to include as many HCW as possible, but the reality was that midwives were the main ones motivated to fill out the survey. Also, in line with the qualitative interviews performed (not shown in this publication), we are aware the midwives are the ones with the closest relationship with pregnant women.

Discussion and conclusion are well written.

Authors’ response: Thank you very much.

Reviewer 4 Report

This study is a research on the acceptance and rejection of the covid 19 vaccine in pregnant women and health personnel.

Among the limitations of the study, the fact that the majority of the participants had a high level of education should also be included. Because those with low education level are almost non-existent in the study. Access to information and acceptance of innovations by people with higher education may be higher, but on the contrary, it may be lower. Therefore, there may be a bias.  Likewise, high gestational age can also be an impressive factor.

2.  In the References section, the names of the authors are either three names followed by et al, or six names followed by et al. Care should be taken to write.

3. There are other studies on covid vaccine rejection in the literature. It is recommended to include them in the reference section as well.

Eg. COVID-19 Vaccine Hesitancy in Healthcare Personnel: A University Hospital Experience. Beril Kara Esen 1, Gunay Can 1, Betul Zehra Pirdal 1, Sumeyye Nur Aydin 1, Aysenur Ozdil 1, Ilker Inanc Balkan 2, Beyhan Budak 2, Yilmaz Keskindemirci 3, Ridvan Karaali 2, Nese Saltoglu 2

Vaccines (Basel) . 2021 Nov 17;9(11):1343. doi: 10.3390/vaccines9111343.

Author Response

This study is a research on the acceptance and rejection of the covid 19 vaccine in pregnant women and health personnel.

Among the limitations of the study, the fact that the majority of the participants had a high level of education should also be included. Because those with low education level are almost non-existent in the study. Access to information and acceptance of innovations by people with higher education may be higher, but on the contrary, it may be lower. Therefore, there may be a bias. Likewise, high gestational age can also be an impressive factor.

Authors’ response: Thank you for the reviewer for realising this potential limitation. However, as discussed with the research team, especially those HCW working at antenatal care, they were not surprised of these results, as they were aware of the high level of education of the pregnant population in the metropolitan area of Barcelona. This might be a bias when compared with all the population of Catalonia, Spain etc., but the representation of the population in this study is representative of the pregnant women in the area, thus, we are confident that there is no selection bias.

  1. In the References section, the names of the authors are either three names followed by et al, or six names followed by et al. Care should be taken to write.

Authors’ response: Thank you for this observation. We have made the references consistent, with one author name followed by et al.

  1. There are other studies on covid vaccine rejection in the literature. It is recommended to include them in the reference section as well.
  • COVID-19 Vaccine Hesitancy in Healthcare Personnel: A University Hospital Experience. Beril Kara Esen 1, Gunay Can 1, Betul Zehra Pirdal 1, Sumeyye Nur Aydin 1, Aysenur Ozdil 1, Ilker Inanc Balkan 2, Beyhan Budak 2, Yilmaz Keskindemirci 3, Ridvan Karaali 2, Nese Saltoglu 2 Vaccines (Basel) . 2021 Nov 17;9(11):1343. doi: 10.3390/vaccines9111343.

Authors’ response: We would like to thank the reviewer for recommending this article that has now been included in the manuscript.